# The Plasma Membrane P-Type ATPase CtpA Is Required for *Mycobacterium tuberculosis* Virulence in Copper-Activated Macrophages in a Mouse Model of Progressive Tuberculosis

**DOI:** 10.3390/biomedicines13020439

**Published:** 2025-02-11

**Authors:** Marcela López-Ruíz, Jorge Barrios-Payán, Milena Maya-Hoyos, Rogelio Hernández-Pando, Marisol Ocampo, Carlos Y. Soto, Dulce Mata-Espinosa

**Affiliations:** 1Chemistry Department, Faculty of Sciences, Universidad Nacional de Colombia, Ciudad Universitaria, Carrera 30 N° 45-03, Bogota 11321, Colombia; gmlopezr@unal.edu.co (M.L.-R.); mmayah@unal.edu.co (M.M.-H.); 2Department of Pathology, Experimental Pathology Section, National Institute of Medical Sciences and Nutrition ‘‘Salvador Zubirán”, Mexico City 14080, Mexico; qcjbp77@yahoo.com.mx (J.B.-P.); rhdezpando@hotmail.com (R.H.-P.); 3Chemistry Department, Faculty of Mathematical and Natural Sciences, Universidad Distrital Francisco José de Caldas, Carrera 3 N° 26A-40, Bogota 110311, Colombia; mocampoc@udistrital.edu.co

**Keywords:** *Mycobacterium tuberculosis*, copper, P_1B_-ATPase CtpA, whole cell attenuated strain, oxidative stress, ROS

## Abstract

**Background/Objective**: Finding new targets to attenuate *Mycobacterium tuberculosis* (*Mtb*) is key in the development of new TB vaccines. In this context, plasma membrane P-type ATPases are relevant for mycobacterial homeostasis and virulence. In this work, we investigate the role of the copper-transporting P-type ATPase CtpA in *Mtb* virulence. **Methods**: The impact of CtpA deletion on *Mtb*’s capacity to overcome redox stress and proliferate in mouse alveolar macrophages (MH-S) was evaluated, as well as its effect on *Mtb* immunogenicity. Moreover, the influence of CtpA on the pathogenicity of *Mtb* in a mouse (BALB/c) model of progressive TB was examined. **Results**: We found that MH-S cells infected with wild-type (*Mtb*H37Rv) or the mutant strain (*Mtb*H37RvΔ*ctpA*) showed no difference in *Mtb* bacterial load. However, the same macrophages under copper activation (50 µM CuSO_4_) showed impaired replication of the mutant strain. Furthermore, the mutant *Mtb*Δ*ctpA* strain showed an inability to control reactive oxygen species (ROS) induced by PMA addition during MH-S infection. These results, together with the high expression of the *Nox2* mRNA observed in MH-S cells infected with the *Mtb*∆*ctpA* strain at 3 and 6 days post-infection, suggest a potential role for CtpA in overcoming redox stress under infection conditions. In addition, *Mtb*Δ*ctpA*-infected BALB/c mice survived longer with significantly lower lung bacterial loads and tissue damage in their lungs than *Mtb*H37Rv-infected mice. **Conclusions**: This suggests that CtpA is involved in *Mtb* virulence and that it may be a target for attenuation.

## 1. Introduction

Tuberculosis (TB) is a public health issue and the leading cause of mortality produced by a single infectious pathogen, known as *Mycobacterium tuberculosis* (*Mtb*) [1]. In 2023, there were 10.8 million new cases and 1.25 million deaths caused by TB [1]. The incidence of TB is primarily attributed to high rates of chemotherapy abandonment associated with the toxicity of certain anti-TB drugs, the emergence of multi- and extensively drug-resistant (MDR and XDR) *Mtb* strains, and co-infection with HIV [2,3]. Unfortunately, the Bacillus Calmette-Guérin (BCG), the only licensed vaccine against TB, demonstrates variable efficacy depending on the age and origin of the immunized population. Specifically, the BCG vaccine is effective in protecting pediatric populations from severe forms of TB; however, its efficacy against pulmonary TB (the main form of the disease) in adolescents and adults is variable [4,5]. Therefore, new strategies to control TB and the development of effective vaccine candidates against all forms of TB are pivotal.

To date, there are sixteen anti-TB vaccine candidates, five of which are currently undergoing Phase III clinical trials. However, thus far, none of the vaccines have demonstrated sufficient capacity to prevent or eradicate TB [1,6,7,8,9]. Among the anti-TB vaccine candidates, those based on live attenuated mycobacteria are of particular interest due to their capacity to display a broad antigenic repertoire and maintain the full complement of T cell epitopes that elicit an effective immune response [10,11].

It is therefore imperative to identify alternative *Mtb* attenuation targets and gain a deeper understanding of the molecular mechanisms that enable *Mtb* to evade macrophages during the infection process and facilitate the development of new TB vaccine candidates [12]. In this sense, membrane proteins are interesting targets for the rational design of anti-TB vaccines, given their pivotal function as transporters and mediators in the pathogen–environment interaction [13]. Several studies have demonstrated the importance of the P-type ATPase plasma membrane transporters for maintaining ionic homeostasis within cells and for the survival of *Mtb* [14,15,16,17,18,19,20].

*Mtb* encodes 12 open reading frames encoding P-type ATPases, classified according to ion specificity and transmembrane topology. Three copper transporters have been identified: CtpA, CtpB, and CtpV. These play an important role in mycobacterial heavy metal homeostasis by transporting cations across the plasma membrane against a concentration gradient [14,21,22].

Some studies have demonstrated that alterations in copper concentrations within the phagosome serve as an antimicrobial mechanism against intracellular pathogens, such as *Mtb* [23,24,25,26]. Moreover, the existence of multiple copper-transporting P-type ATPases within the mycobacteria indicates the potential for diverse physiological functions, including the regulation of cytoplasmic metal levels (detoxification) and the metalation of membrane/periplasmic cuproenzymes (alternative role) [15,16,20,23,27,28,29].

Prior research has indicated that P_1B_-type ATPases (heavy metal transporters) of *Mtb* with alternative functions, in contrast to their homologous counterparts (metal detoxification), may be essential for survival in the harsh environmental conditions of the phagosome and for the persistence of *Mtb* in vivo infection models [20,28].

Among the three *Mtb* copper-transporting P-type ATPases, only CtpV has been demonstrated to be essential for the detoxification of bacilli from copper [30,31]. The expression of *ctpV* is stimulated by high concentrations of copper and is controlled by *cso*, a copper-sensitive operon of *Mtb* [30]. *Mtb*Δ*ctpV* mutant cells are unable to tolerate toxic doses of copper. Furthermore, the absence of *ctpV* has been demonstrated to result in elevated intracellular copper levels [30,31]. However, the deletion of *ctpV* in *Mtb* did not result in a significant reduction in colonization levels in a murine model [31].

Our research group has been interested in the relationship between bacterial copper response and the virulence of *Mtb*. This research has led to the characterization and study of CtpB and CtpA ATPases expressions under stress conditions that mimic the intra-phagosomal environment [22]. Our initial findings demonstrated that CtpB and CtpA are involved in the pumping of copper (I) from mycobacterial cells [14,15]. Furthermore, the disruption of *ctpB* in *Mtb* did not result in the susceptibility of *Mtb*Δ*ctpB* cells to toxic doses of copper, nor did it lead to the accumulation of intracellular copper in cells cultured under high concentrations of copper. These findings suggest that CtpB is not directly associated with copper detoxification, but rather, it could play an alternative role in copper transport [15].

On the other hand, the expression of *ctpA* was observed to increase during infection, indicating that the function of this copper transporter is essential for the virulence of *Mtb* [19,32]. Additionally, *ctpA* is induced in low Cu^2+^ concentrations and under oxidative and nitrosative conditions in vitro. However, this upregulation was not observed in high concentrations of Cu^2+^ [16,21]. Furthermore, it is notable that the growth rate of the wild-type and *ctpA* mutant strain (*Mtb*∆*ctpA*) was comparable at both low and high doses of Cu^2+^. Additionally, disruption of *ctpA* did not result in intracellular copper accumulation, indicating that copper-mediated transport by CtpA may have an alternative function [16,21]. However, the *Mtb*∆*ctpA* strain exhibited a diminished capacity for growth when exposed to oxidative and nitrosative stress in vitro. To this end, we sought to determine whether a possible correlation exists between Cu(I) transport mediated by CtpA and the activity of MmcO, which, in turn, mediates the in vitro oxidative capacity of the phenolic compounds ABTS and pPD in lysates of mycobacterial strains [33,34]. This investigation revealed that whole-cell lysates of the *Mtb*∆*ctpA* strain, previously stimulated with Cu^2+^, H_2_O_2_, and sodium nitroprusside, exhibited a diminished capacity to oxidize organic substrates. These findings suggest that the efflux of copper transported by CtpA plays a crucial role in overcoming redox stress and may be essential for metalation and activity of MmcO in *Mtb* [16]. However, further investigation is required to determine the extent to which CtpA contributes to the virulence of *Mtb*.

In this context, the present study observed that a deletion of *ctpA* affects intracellular *Mtb* replication in alveolar macrophages (MH-S) under copper activation during the assay. Furthermore, a comparative analysis of BALB/c mice infected with the *Mtb*Δ*ctpA* strain revealed that these animals exhibited prolonged survival times, a reduced bacillary load, and a lower percentage of pneumonia in comparison to animals infected with the parental *Mtb*H37Rv strain. These findings collectively indicate that CtpA plays a pivotal role in the survival and virulence of *Mtb*. Consequently, this plasma membrane transporter may be regarded as a prospective target for attenuation in the development of a new TB vaccine.

## 2. Materials and Methods

### 2.1. Bacterial Strains and Growth Conditions

Bacterial strains used in this study are shown in Appendix A. The *ctpA* mutant strain of *Mtb*H37Rv was obtained using the recombineering technique with the Chec9c system, which promoted the allelic exchange between *ctpA* in bacterial DNA and the encoding gene for antibiotic resistance in the allelic exchange substrate (AES) [35]. The AES of *ctpA* [16] was cloned in the recombineering strain *Mtb*H37Rv:pJV53 [11], and colonies with defective *ctpA* (*Mtb*Δ*ctpA*) were selected for hygromycin (Hyg) resistance on 7H10 plates.

*Mtb* strains were cultured in Middlebrook 7H9 broth (Catalog 271310, BD Biosciences, Wokingham, Berkshire, UK) with supplements (10% OADC, 0.5% glycerol, and 0.05% tyloxapol (Sigma-Aldrich, St. Louis, MO, USA) at 35 °C with gentle agitation (70 rpm) until reaching mid-log phase measured at 600 nm (1.0–1.3), or on Middlebrook 7H10 and 7H11 agar (Catalog 262710 and 212203, respectively, BD Biosciences, Wokingham, Berkshire, UK) supplemented with OADC and 0.5% glycerol.

Kinetics of mycobacterial growth were conducted according to the procedures reported previously by our laboratory [11]. Isolated colonies of *Mtb*Δ*ctpA* and *Mtb*H37Rv were cultured in 7H9 broth until reaching the stationary phase (OD_600_ ~ 2.0–2.5, for approximately 30 days). The OD_600_ of the mycobacterial cultures was then measured daily in a spectrophotometer (Thermo Scientific, Waltham, MA, USA) [11]. For the infection experiments (Figure 1), batches of the mycobacterial strains *Mtb*H37Rv and *Mtb*Δ*ctpA* were harvested during the mid-log phase, counted, aliquoted, and stored at −80 °C until use.

### 2.2. MH-S Cells Culture

Murine alveolar macrophage MH-S cell line (ATCC CRL-2019) was cultured in a 10 cm^2^ cell culture dish containing Roswell Park Memorial Institute Medium (RPMI) (Gibco^TM^, Waltham, MA, USA) with 2 mM l-glutamine, 10 mM HEPES, 2.0 g/L NaHCO_3_, pH 7.4) supplemented with 10% fetal bovine serum (FBS) (medium complete) and incubated at 37 °C in 5% CO_2_ atmosphere until reaching 80% confluence.

### 2.3. Infection of MH-S Cells with Mtb Strains

1 × 10^5^ cells were co-cultured with *Mtb* strains at an MOI (Multiplicity of infection) of 1:4 and incubated for one hour at 37 °C and 5% CO_2_ to allow the infection. To remove extracellular bacteria, three washes with RPMI medium plus 1% penicillin and 1% streptomycin (Gibco^TM^, Waltham, MA, USA) were performed. Then, cells were carefully resuspended in 1 mL RPMI-10% SFB medium with or without CuSO_4_, and incubated for 1, 2, 3, or 6 days. At each time post-infection, the supernatants were harvested by centrifugation at 14,000 rpm for 10 min and stored at −80 °C until further analysis (for cytokine and nitric oxide determinations). The infected MH-S cells were lysed with 0.1% SDS and neutralized with 20% BSA in 7H9 broth. The viability of intracellular bacteria was determined by counting colony-forming units (CFU) through serial dilution method after 21 days of incubation at 37 °C in 5% CO_2_.

### 2.4. Cytokine Production in Mtb-Infected MH-S Cells

TNF-α, and IL-12 concentrations in infected MH-S cell culture supernatants were quantified using a sandwich ELISA with the BD OptEIA^TM^ Mouse IL-12 (p40) ELISA Set (Cat. No. 555165), and the BD OptEIATM Mouse TNF-α ELISA Set II (Cat. No. 558534). Assays were conducted in accordance with the manufacturer’s instructions and our laboratory specifications [11].

### 2.5. Determination of the Nitric Oxide (NO) Concentration Produced by Infected MH-S

The nitrite concentration was used as a measure of NO production [36]. To quantify the nitrite concentration, 50 μL of the previously supernatants collected was reacted with 150 μL of Griess reagent (0.2% *N*-(1-Naphthyl) ethylenediamine dihydrochloride, and 2% sulfanilamide in 5% phosphoric acid) for 30 min at 35 °C. Absorbance was measured at 540 nm. A calibration curve was prepared in parallel using NaNO_2_ (0–27.5 ng/μL) in triplicate in the 96-well plate, to which an equivalent volume of Griess reagent was added.

### 2.6. RNA Extraction and cDNA Synthesis from Mtb-Infected Cells

RNA was extracted from MH-S macrophages infected with mycobacteria using the Quick-Start Protocol RNAeasy^®^ Mini Kit (Cat. No. 74104, Qiagen, Germantown, MD, USA) according to the manufacturer’s instructions. cDNA libraries were constructed with 0.5 μg RNA, 1 μL Oligo(dT)12–18, 1 μL dNTP mix, and RNase-free water, with the mouse leukemia virus reverse transcriptase (M-MLVRT, Cat. No. 28025013; Invitrogen™, Carlsbad, CA, USA). The concentration of cDNA and the Abs 260/280 ratio were determined using an Epoch Microplate Reader (Bio Tek Instruments, Inc., Winooski, VT, USA).

### 2.7. RT-qPCR in Mtb-Infected Macrophages

The transcription levels of mRNA encoding *IL-1β*, *Nos2*, and *Nox2* were evaluated by RT-qPCR in MH-S cells infected with the mycobacterial strains. Quantitative PCR (qPCR) reactions with 100 ng of cDNA were measured using SsoAdvanced Universal SYBR^®^ Green Supermix in a CFX-96 thermal cycler (Biorad, CA, USA). The specific details of the qPCR program and oligonucleotide sequences [37,38,39] are provided in Appendix A. Efficiency curves were constructed using cDNA serial dilutions (0.01–100 ng/μL) from uninfected MH-S macrophages (Appendix A). The relative transcription levels were calculated using the Pfaffl method [40], with glyceraldehyde-3-phosphate dehydrogenase (*gapdh* mRNA) as the reference gene. Each quantification was performed in triplicate, with a negative control reaction lacking cDNA.

### 2.8. Quantification of Reactive Oxygen Species (ROS) Produced in Mtb-Infected Macrophages

The concentration of ROS in MH-S macrophages infected with mycobacteria was determined using the NBT assay for measuring intracellular superoxide (O_2_^•–^) production [41,42]. 1.0 × 10^5^ MH-S cells were seeded into 12-well dishes in RPMI-10%FBS. The cells were incubated at 37 °C and 5% CO_2_ for 48 h. Then, the cells were infected with *Mtb* strains (MOI 1:5) and incubated for an additional hour at 37 °C and 5.0% CO_2_; uninfected cells were included in the assay as a control for ROS production. Subsequently, the supernatants were removed, and each well was washed three times with 1X PBS and a 1% antibiotic solution. Subsequently, 300 μL of each of the solutions—Y-NBT (1 mg/mL), Y-NBT with PMA (900 ng/mL), and Y-NBT with PMA (900 ng/mL) and SOD (30 μg/mL), all prepared in DMEM/F-12 without phenol red (Gibco™ 21041025, Waltham, MA, USA)—were added separately to the wells in triplicate. The plates were incubated for 90 min at 37 °C with 5% CO_2_. The cells were washed three times with warm PBS, twice with methanol, and then air dried. The NBT reduction product was dissolved in 240 µL of KOH (2 M) and 280 µL of DMSO (100%), and the plates were incubated at 35 °C with shaking for 10 min. 100 µL of the supernatants (in quadruplicate) were transferred to a 96-well plate and the absorbances were determined at 620 nm in an Epoch microplate Reader (Bio Tek Instruments, Inc., Winooski, USA). To quantify the NBT dissolution, a calibration curve was prepared with NBT (0.312–20 nmol) in a solution of 1:1.2 (KOH:DMSO), as previously described by Rook et al. [42].

### 2.9. Progressive Pulmonary TB Model of BALB/c Mice

The animals used in the infection model were provided by the Department of Experimental Research and Animal Housing of the INCMNSZ. For the infection, aliquots of the different *Mtb* strains were thawed and sonicated for 45 s to prevent the formation of clumps formations. For each *Mtb* strain (*Mtb*H37Rv, and *Mtb*H37RvΔ*ctpA*), a total of 16 mice (8-week-old BALB/c mice with an average weight of 22 g) were assigned to each group. Mice were anesthetized with sevoflurane (Sigma-Aldrich, St. Louis, MO, USA) and infected endotracheally with 2.5 × 10^5^ CFU in 100 μL of PBS [43]. Following infection, mice were housed in boxes with HEPA-filtered ventilation (Lab Products, model Super Mouse 1800™ AllerZone™ Micro-Isolator^®^, LabProducts Inc., Houston, TX, USA) with 12 h light-dark cycles. After animal infection, the remnant of the bacterial inoculum was plated to confirm the number of CFU administered to the animals. If, during the evolution of the experiment, animals exhibited poor mobility, piloerection, high respiratory frequency, or a weight loss > 20%, the humane endpoint criteria were applied.

On days 3, 21, 60, and 120 post-infection, groups of four mice were euthanized by exsanguination under anesthesia with sodium pentobarbital (Sigma-Aldrich, St. Louis, MO, USA) at a dose of 210 mg/kg. The right lungs were promptly frozen, and were stored at −80 °C until processing. To estimate the bacterial burdens, lungs were homogenized, and the bacillary loads were determined by counting CFU through serial dilution method after 21 days of incubation at 37 °C in 5% CO_2_. The left lungs were perfused with ethanol, fixed for 72 h, and embedded in paraffin for further histological analysis. Photomicrographs were taken at 25× on a Leica microscope and pneumonic areas were delineated and measured using the automated Leica Application Suite v4.0 software (LASv4.0), and the percentage of lung surface affected by pneumonia was calculated. Survival curves were constructed with the animals used in virulence kinetics using GraphPad Prism 9 (GraphPad Software, San Diego, CA, USA).

### 2.10. Statistical Analysis

All statistical analyses were performed using GraphPad Prism 9 Software. Data were analyzed using Student’s t-comparison test to evaluate the CFU/mL, RT-qPCR, ROS and cytokines concentration results between strains. Differences of *p* < 0.05 (*), *p* < 0.01 (**), *p* < 0.001 (***), and *p* < 0.0001 (****) were considered to be statistically significant.

## 3. Results

### 3.1. The ctpA Deletion Does Not Alter the Mtb Growth Kinetics in Standard Culture Conditions

Prior to conducting the infection assays, the growth curves of *Mtb*H37RvΔ*ctpA* and *Mtb*H37Rv under standard culture conditions were obtained (Figure 2). The time and OD_600_ corresponding to the mid-exponential growth phase were determined (Log_1/2_).

The mutant strain reached the Log_1/2_ phase at OD_600_ of 1.028, which is comparable to the previously reported value of 1.07 for *Mtb*H37Rv under identical conditions [11]. Furthermore, as demonstrated in Figure 2, the *Mtb*H37RvΔ*ctpA* strain exhibited a shorter latency phase in comparison with the parental strain. The replication rate of the mutant (0.105 OD_600_/day) in the log phase was found to be lower than the parental strain (0.192 OD_600_/day) [11]. This indicates that the *ctpA* gene is not indispensable for the growth rate of *Mtb* in standard laboratory culture conditions. However, it may exert an influence on the bacilli adaptability and replication rate.

### 3.2. ctpA Is Required for Intracellular Proliferation of Mtb in Copper-Activated Alveolar Macrophages

To ascertain whether, as with other P_1B_-type ATPases, CtpA may be involved in the intracellular survival of *Mtb* [20,28], the bacterial load (CFU/mL) was quantified at different post-infection times in mouse alveolar macrophages (MH-S cell line) infected with the wild-type *Mtb*H37Rv or mutant *Mtb*Δ*ctpA* strain at MOI 1:3 (Figure 3). Considering that CtpA is a Cu^+^-ATPase, it is possible that the bacterial growth could be affected by the presence of copper, which is required for the activation of CtpA [16]. To investigate this, the bacterial load was assessed with and without 50 μM CuSO_4_ (Figure 3).

As shown in Figure 3, at 1 h post-infection (hpi), which is the time period at which mycobacteria are predominantly phagocytosed [44,45], the mutant strain *Mtb*Δ*ctpA* exhibited a diminished capacity to infect MH-S cells, in comparison to the parental strain, and independent of the presence of copper (*p* < 0.05). However, in the absence of copper during late infection, no differences were observed in the replication of the *Mtb*Δ*ctpA* strain in comparison to the wild-type strain. The *Mtb*H37Rv strain exhibited 138-fold increase (1 h and 6 dpi: 4300 and 594,286 CFU/mL, respectively) in colony-forming units (CFU) at the sixth day post-infection, while the mutant strain showed a 136-fold increase (1 h and 6 dpi: 2550 and 416,250 CFU/mL, respectively) in the same time period (Figure 3). In contrast, when the MH-S infection advanced in the presence of CuSO_4_ (50 μM), the *Mtb*Δ*ctpA* mutant displayed impaired proliferation, resulting in a significantly reduced number of bacteria (CFU/mL) from one to six days post-infection compared with the control strain (Figure 3). The *Mtb*H37Rv strain displayed a 11.5-fold increase in CFU during the 6 dpi period (with 50 μM CuSO4; 1 h and 6 dpi: 20,091 and 231,333 CFU/mL, respectively), whereas the *Mtb*Δ*ctpA* strain exhibited a 6.22-fold increase at the same infection time (with 1 h and 6 dpi: 5,389 and 33,500 CFU/mL, respectively). These findings indicate that, as observed in other murine macrophage cell lines, copper enhances the bactericidal activity of alveolar macrophages of the MH-S line, thereby contributing to the control of *Mtb* infection [46,47]. These findings also indicate that *ctpA* is necessary for the optimal survival and intracellular multiplication of *Mtb* within MH-S macrophages in the presence of 50 μM CuSO_4_ (Figure 3). Therefore, it can be suggested that the intracellular proliferation of *Mtb* is impaired by *ctpA* deletion, which is probably due to the specific effect of macrophage activation by copper.

### 3.3. Copper Modulates the Proinflammatory Response of MtbΔctpA-Infected MH-S Cells

The objective of developing live attenuated vaccines against mycobacteria is to provide activation to the immune system similar to the natural infection, with minimal or no immunopathology [48]. Considering the low intracellular proliferation of *Mtb*Δ*ctpA* mutant in MH-S cells, we assessed whether the *ctpA* deletion affects the production of proinflammatory cytokines during infection of mouse alveolar macrophages. Consequently, we quantified the relative expression of *IL-1β* mRNA, as well as the levels of interleukin-12 (IL-12) and TNF-α produced by MH-S cells infected with *Mtb* strains in the presence and absence of CuSO_4_ (Figure 4 and Figure 5).

Gene expression analysis in MH-S cells shows that cells infected with the *Mtb*Δ*ctpA* strain in the absence of CuSO_4_ exhibited a higher level of *IL-1β* transcription than cells infected with the wild-type strain at 6 dpi, which is not observed in the presence of the metal (Figure 4). Similarly, the *Mtb*Δ*ctpA* strain induced statistically significantly higher levels of TNF-α and IL-12 than the control strain at 3 to 6 dpi, only in the absence of the metal (Figure 5). These findings suggest that, in the absence of copper *ctpA*, deletion affects the immunogenicity of *Mtb* in vitro; however, the infection of MH-S cells with *Mtb*Δ*ctpA* induces the production of key protective cytokines that contribute to the control of bacilli growth by macrophages [49,50,51,52].

Furthermore, our findings suggest that CuSO_4_ (50 μM) may modulate the host proinflammatory response [53] of macrophages infected with the *Mtb*Δ*ctpA* mutant strain. At 6 dpi, the presence of copper during the assay resulted in a significant reduction in *IL-1β* mRNA and proinflammatory cytokines (TNF-α and IL-12) produced in response to *Mtb*Δ*ctpA* infection, as compared to the levels observed at the same period of infection in the absence of the metal. Conversely, in the presence of CuSO_4_, macrophages infected with the parental strain showed a less pronounced reduction in IL-1β mRNA gene expression and IL-12 production, and an increased level of TNF-α (Figure 4 and Figure 5) at 6 dpi.

### 3.4. MtbΔctpA Mutant Displays Impaired Ability to Control Reactive Oxygen Species (ROS) During Infection of MH-S Cells

In a previous report, we discussed the potential relevance of CtpA-mediated copper transport in the ability of the bacillus to cope with redox stress under in vitro conditions [16]. To evaluate whether the deletion of *ctpA* affects the redox stress exerted by macrophages under infection, the production of ROS (O_2_^•−^) and RNS (NO) was compared in MH-S cells infected with *Mtb*H37Rv or *Mtb*Δ*ctpA*. Additionally, the expression of the genes encoding key enzymes in the production of NO (iNOS*,* the *Nos2* gene) and O_2_^•−^ (NADPH oxidase, the *Nox2* gene) was determined [54,55,56,57]. Activation of the *Nox2* gene was higher (3.41-fold) at day 6 in MH-S cells infected with the mutant strain *Mtb*Δ*ctpA* compared to cells infected with the wild strain *Mtb*H37Rv (Figure 6a). Moreover, stimulation of macrophages with 50 μM CuSO_4_ resulted in activation of the *Nox2* gene with the *Mtb*Δ*ctpA* mutant strain from 3 dpi (Figure 6a). Conversely, transcription of the *Nos2* (*iNOS*) mRNA and NO production exhibited a time-dependent induction in *Mtb*-infected macrophages (Figure 6b and Figure 7). However, there were no significant differences in NO production between cells infected with the wild-type or the mutant strain.

Furthermore, as demonstrated in Figure 8, the addition of the inducing agent PMA [55] (dotted bars) resulted in a higher ROS production in macrophages infected with the mutant strain (*Mtb*Δ*ctpA*) compared to the production obtained in the absence of the inducer (solid bars). These findings align with those observed in the positive control group (uninfected macrophages) [55]. In contrast, the intracellular ROS concentration in cells infected with the wild-type strain is observed to decrease even after stimulation with PMA. This suggests that the deletion of *ctpA* impairs the capacity of *Mtb* to scavenge ROS generated within infected macrophages [58].

### 3.5. MtbΔctpA Strain Attenuates Virulence in Mice

To evaluate the level of virulence in vivo of the *ctpA* deletion in *Mtb*, the survival, bacterial loads, and extension of tissue damage (percentage of lung area affected by pneumonia) were determined in BALB/c mice infected intratracheally with 2.40 × 10^5^ CFU of *Mtb*H37Rv or 2.07 × 10⁵ CFU *Mtb*H37RvΔ*ctpA* [43,48,59]. Almost all of the animals infected with the Δ*ctpA* strain survived during the whole experimental period (four months), with only one animal dying at 84 dpi. In contrast, mice infected with the parental strain started to die at 49 dpi and all the animals died at 65 dpi (Figure 9a). These results correlated well with those determined by CFUs of lung homogenates. The bacilli inoculum was similar for both strains; however, at 3 dpi, the number of CFU in the lungs of mice infected with the mutant *Mtb*H37RvΔ*ctpA* was so low that it was below the detection limit (100 CFU/lung). In contrast, in the lungs of animals infected with the parental strain, approximately 3.93 × 10^5^ CFU/lung was observed (Figure 9b). Furthermore, with respect to the bacilli inoculum, at 21 dpi, the bacterial load in the lungs of mice infected with the wild-type and mutant strain exhibited an increase of approximately 5.46-fold (1.31 × 10^6^ vs. 2.40 × 10^5^ CFU) and 1.44-fold (2.98 × 10^5^ vs. 2.07 × 10^5^ CFU), respectively. At 60 dpi, the bacterial load exhibited an additional increase of ~20-fold (2.59 × 10^7^ CFU) and ~2.57-fold (7.65 × 10^5^ CFU) in the tissues of animals infected with the wild-type and mutant strains, respectively (Figure 9b). By day 120, animals infected with the *Mtb*Δ*ctpA* mutant strain exhibited a similar bacterial load (4.79 × 10^5^), while no survival animals infected with the parental strain were available at this time point (Figure 9a).

The quantification of pneumonia in animals infected with the parental strain at 60 dpi collectively revealed approximately 40% of lung consolidation (Figure 9b). The remaining tissue exhibited extensive regions of alveolitis and diffuse alveolar damage, as well as perivascular inflammatory infiltrate, predominantly composed by lymphocytes (Figure 10a and Appendix A). The extensive tissue damage and high bacilli burdens caused by the parental strain resulted in total mortality before reaching the final euthanasia time point on day 120 (Figure 9a). In contrast, the *Mtb*Δ*ctpA* at 60 dpi produced few and small patches of pneumonia that collectively represented less than 10% of the total lung area (Figure 10 and Appendix A). The areas of alveolitis and the perivascular infiltrates were scarce. By day 120 post-infection, the pneumonic areas were well delimited and affected approximately 20% of the total lung surface, and the areas of alveolitis were minimal, as well as the perivascular infiltrates (Figure 10a and Appendix A). The reduction in histological damage and the low bacterial burden of the mutant strain in comparison with the parental strain indicates that CtpA plays a significant role in the virulence of mycobacteria (Figure 10).

## 4. Discussion

As has been previously reported, P_1B_-ATPase metal transporters have been recently identified as a relevant virulence factor for several intracellular pathogens, including *Mtb* [20,28,31,60]. The existence of multiple coding genes for these transporters with identical metal specificity and transport direction, suggest that they perform non-redundant roles during infection, which could be crucial for survival in hostile environmental conditions [20,27,28,29,61]. A recent study indicated that CtpA may have an alternative function distinct from copper detoxification. Thus, CtpA appears to be essential for the metalation of secreted proteins and the overcoming of redox stress [16]. Nevertheless, the precise role of this transporter during infection remains unknown. To determine the potential role of CtpA during *Mtb*-infection, the effect of the *ctpA* gene deletion on *Mtb*H37Rv virulence was evaluated in a BALB/c mouse model of progressive pulmonary TB [59].

The results of this work confirm that the Δ*ctpA* mutant strain is markedly attenuated, enabling the complete survival of infected animals following a four-month infection. Moreover, the bacillary loads in the lungs of infected animals were significantly lower than those observed in animals infected with the parental strain. As early as 3 dpi, the number of live bacilli in the infected lungs was below the limits of detection, clearly indicating that the *ctpA* deletion resulted in an early and deeply impaired *Mtb* growth proliferation, in contrast to mice infected with the wild-type *Mtb*H37Rv parental strain, which produced progressive bacillary proliferation in the lungs [59]. Meanwhile, the high bacillary load in the *Mtb*H37Rv-infected animals at 65 dpi (7.41 log_10_ CFU/lung), co-existed with extensive tissue damage and animal death. At 21 dpi and a later stage (120 dpi), the lungs of mutant-infected mice showed similar bacilli burdens (5.68 log_10_ CFU/lung). These findings suggest that the *ctpA* mutation plays a significant role during the initial stages of *Mtb* infection, which may be responsible for the diminished bacterial replication capacity and subsequent attenuation of *Mtb*Δ*ctpA*. Consequently, the murine model of pulmonary TB indicates that the deletion of *ctpA* in *Mtb* may impact the early immune response, which is crucial for effective host control during the advanced stages of the disease [12]. These observations are consistent with our previous hypothesis that CtpA may play a role mainly in the early stages of the mycobacterial infection process [16]. Our previous report indicated that the transcription of *ctpA* was predominantly induced in *Mtb*H37Ra cultures that were supplemented with concentrations of Cu^2+^ comparable to those found in phagosomes at 24 h post-infection with *Mtb* [16,45].

Given that the attenuation of the *Mtb*Δ*ctpA* strain during the early phase of infection may have been caused by a reduction in its abilities to multiply and/or persist inside the host phagocytic cells, we conducted an examination of the intracellular proliferation of the *ctpA*-deficient mutant in resting and copper-activated alveolar macrophages. In this context, the attenuation exhibited by the *Mtb*Δ*ctpA* mutant strain may be regulated by the presence of copper. The Δ*ctpA* strain did not exhibit any significant growth restriction within resting macrophages. However, it was impaired within macrophages only in the presence of CuSO_4_ 50 μM. Nevertheless, infection of MH-S macrophages with the *Mtb*Δ*ctpA* mutant resulted in the transcription or production of proinflammatory cytokines and microbicidal components essential for the control of *Mtb* during natural infection, such as IL-1β, IL-12, TNF-α, iNOS, and NADPH oxidase [39,43]. The production of IL-1β and TNF-α was also influenced by the presence of copper during infection with the *Mtb*Δ*ctpA* mutant strain. This may prevent the generation of a widespread chronic pro-inflammatory response. The absence or overproduction of certain proinflammatory cytokines, such as IL-1, during *Mtb* infection can result in disease susceptibility and increased host mortality [57,62,63]. Furthermore, the absence or low levels of TNF-α have been linked to a fatal progression of TB, largely due to a reduction in antimycobacterial responses of macrophages and impaired granuloma functionality [57]. An excess of TNF-α has been demonstrated to favor immunopathology by interfering with cell death processes and inducing an hyperinflammatory response [12,62,64]. Therefore, our observations suggest that, in conjunction with changes in copper homeostasis that occur as part of the control mechanisms of *Mtb* infection, a mutation in *ctpA* could result in diverse effects on the magnitude and type of immune response triggered against *Mtb* [12,53]. This may have contributed to the reduction in the immunopathology that occurs during a natural *Mtb* infection, thereby facilitating the attenuation of the mutant strain and the establishment of an effective immune response [9,12,57,65].

The pivotal and alternative roles of *Mtb* P_1B_-type ATPases in *Mtb* persistence are based on their interference with the mechanisms through which *Mtb* faces the redox-hostile conditions in the phagosome [20,28,29]. Considering these observations, our objective was to investigate whether CtpA plays a role in the mechanism by which *Mtb* responds to redox stress during infection. Thus, the *Mtb*Δ*ctpA* mutant strain exhibited increased *Nox2* gene expression at 3 and 6 dpi, in comparison to the wild-type strain. Moreover, if the reduction in NBT is a proportional indicator of intracellular O_2_^•–^ concentration, and its decrease is associated with the ability of the bacillus to block oxygen-dependent microbicidal activity in infected cells [58], *ctpA*-deletion impairs the capacity of *Mtb* to scavenge ROS produced in infected macrophages. *Mtb*Δ*ctpA* mutant is unable to neutralize ROS produced during infection of MH-S alveolar macrophages stimulated with PMA. This result, in conjunction with the previously reported sensitivity to the presence of redox agents in vitro [16], and the diminished capacity to evade ROS accumulation induced by external agents [66] in the absence of *ctpA* in *Mtb* (Appendix A), suggest that CtpA plays a role in the mechanisms that overcome redox stress under infection conditions. As previously reported, the copper transported by CtpA could be relevant for the metalation of the *Mtb*-cuproenzyme, MmcO [16]. Considering the role of MmcO in the scavenging of ROS produced by activated macrophages [34], it can be hypothesized that CtpA could participate in the detoxification of NADPH oxidase-derived ROS through the activation of redox enzymes that require the copper transported by CtpA for their activity [16].

New vaccine strategies should not only consider the ability to generate components of the host immune response with a protective role, such as INF-γ, TNF-α or CD4+ Th1 cells, but also contribute to the generation of a protective host response by overcoming *Mtb* immune evasion mechanisms, restoring the antimicrobial capacity of myeloid cells and their functional interactions with the adaptive immune system [12,65]. It has been proposed that protection at the level of phagocytic function may be achieved by enhancing autophagy, LC3-associated phagocytosis, phagosome-lysosome fusion, ROS production, nutritional immunity, optimal metabolic responses, or antigen presentation and communication with innate and adaptive lymphocytes [12,67]. It has been demonstrated that, by blocking the acquisition of NADPH oxidase and neutralizing host-derived ROS, *Mtb* impairs autophagy-related pathways (LC3) and limits TNF-mediated apoptosis [12,68,69,70]. Considering these findings, it can be postulated that CtpA plays a role in detoxifying ROS, thereby facilitating *Mtb* evasion mechanisms during infection. Consequently, its deletion may enhance the host cell’s redox stress response, including autophagy and programmed cell death, which could, in turn, promote the generation of protective immune responses, which would be the subject of further research.

## 5. Conclusions

In conclusion, the obtained results collectively reinforce interest in the potential of *ctpA* as an attenuation target in the development of an anti-TB vaccine. The attenuation of the *Mtb*H37RvΔ*ctpA* strain in *Mtb* infection indicates the existence of a potential defect in the early phase of infection, which may be associated with a reduced capacity of *Mtb* to proliferate and persist within copper-activated phagocytic cells, accompanied by diminished responsiveness to oxidative stress. Therefore, the deletion of *ctpA* may induce additional control processes in phagocytic cells associated with ROS production, thereby interfering with the bacterial immune evasion mechanisms. Consequently, immunization with a live strain defective in *ctpA* could generate more efficient and prolonged immune responses for bacillus control and host protection.

## Figures and Tables

**Figure 1 biomedicines-13-00439-f001:**
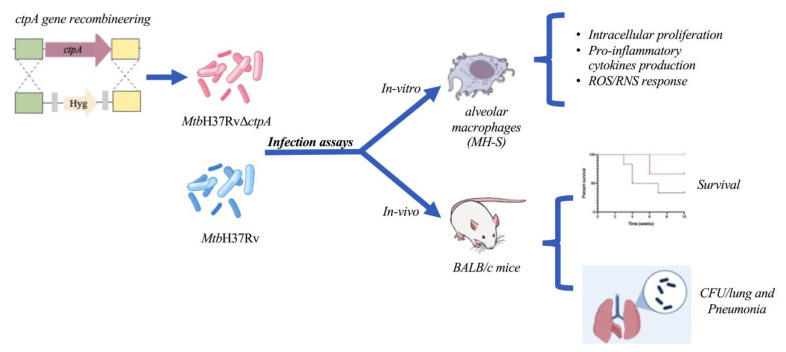
Scheme of the experimental strategy. The deletion of *ctpA* in *Mtb* is evaluated in in vitro and in vivo infection models.

**Figure 2 biomedicines-13-00439-f002:**
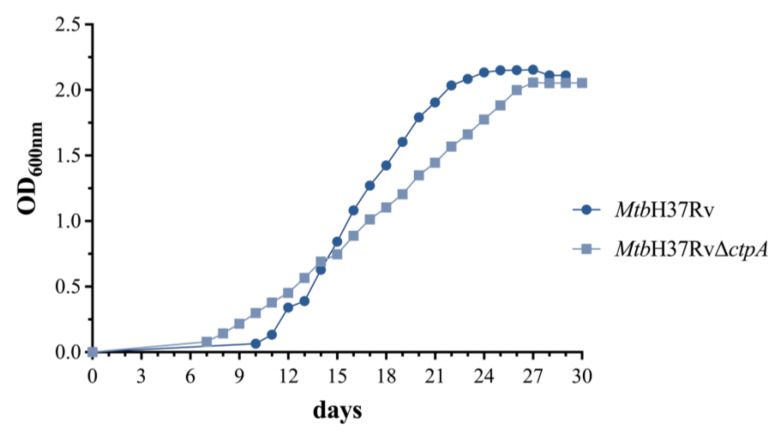
Growth curve of MtbH37Rv and MtbH37RvΔctpA strains in standard cultures*. Mtb* strains were grown in 7H9-OADC-0.05% tyloxapol medium at 37 °C with shaking (70 rpm) in 5% CO_2_. OD_600_ was monitored until the culture reached the stationary phase. Curves represent the mean ± standard deviation of three technical replicates.

**Figure 3 biomedicines-13-00439-f003:**
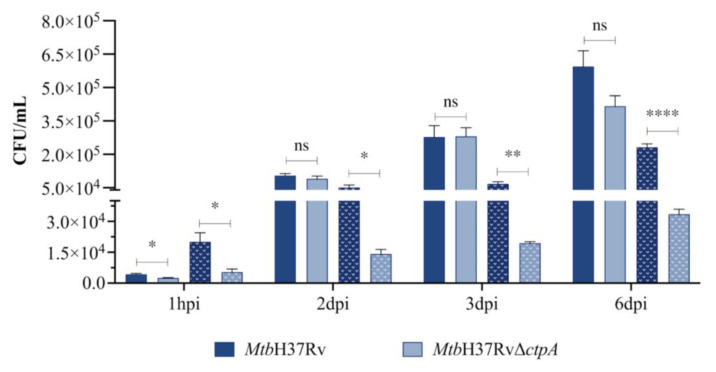
Intracellular proliferation of MtbH37Rv and MtbΔctpA in alveolar macrophages. 1 × 10^5^ macrophages (MH-S) were infected at a 1:3 MOI with *Mtb* strains in the absence (solid bar) or presence of 50 μM CuSO_4_ (dotted bar). Values correspond to the mean ± SEM of three technical replicates from two independent experiments. Asterisks indicate significance (two-tailed Student’s t-comparison unpaired test) at same culture condition (ns *p* > 0.05, * *p* < 0.05, ** *p* < 0.01, **** *p* < 0.0001).

**Figure 4 biomedicines-13-00439-f004:**
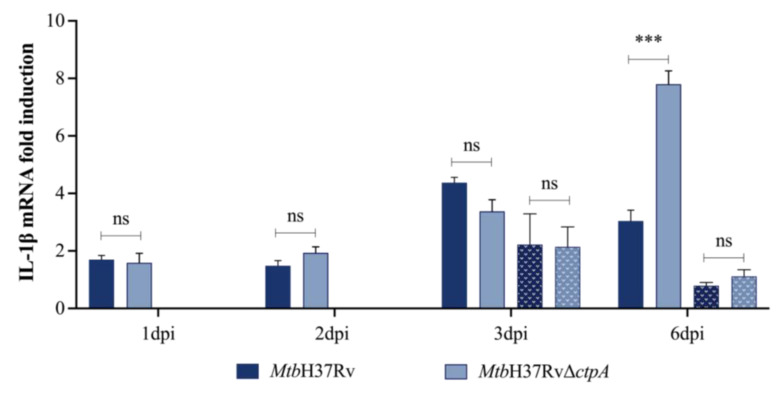
Transcription of IL-1β mRNA in Mtb-infected macrophages. The levels of *IL-1β* mRNA produced by MH-S cells infected with *Mtb*H37Rv or *Mtb*H37RvΔ*ctpA* in the absence (solid bar) or presence of 50 μM CuSO_4_ (dotted bar) were quantified by RT-qPCR. Relative transcription levels were calculated using the Pfaffl method concerning uninfected cells using the *gapdh* gene as a housekeeping gene. Presented values correspond to the mean ± SEM of triplicate of four independent experiments. Asterisks indicate a statistically significant difference with respect to the transcription induced by parental strain infection (ns *p* > 0.5, *** *p* < 0.005).

**Figure 5 biomedicines-13-00439-f005:**
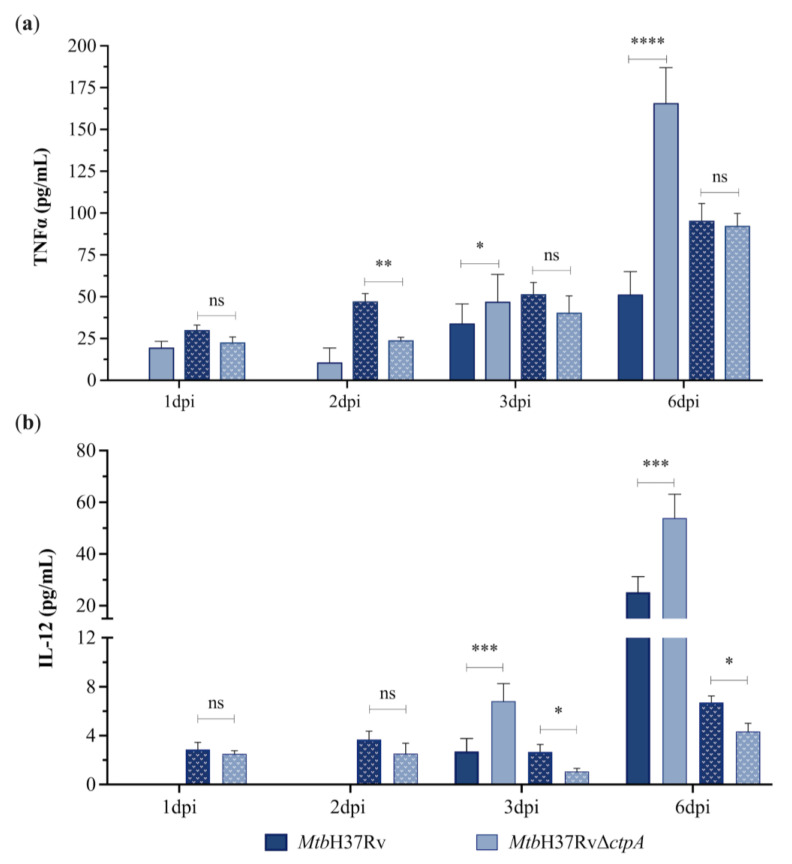
Proinflammatory cytokine secretion by infected macrophages with Mtb strains. The levels of tumor necrosis factor (TNF-α) (**a**) and interleukin-12 (IL-12) (**b**) were determined in the supernatants of MH-S cells infected with *Mtb*H37Rv or *Mtb*H37RvΔ*ctpA* strains at an MOI of 1:3 in absence (solid bar) or presence of 50 μM CuSO_4_ (dotted bar). The plotted values represent the mean concentration (in pg/mL) ± SEM of each cytokine from three independent experiments. Asterisks indicate statistical significance (two-tailed unpaired *t*-test comparison, ns *p* > 0.05; * *p* < 0.05; ** *p* < 0.01;*** *p* <0.001; **** *p* < 0.0001). IL-12 values on days 1 and 2 dpi in the assay without copper are below the detection limit.

**Figure 6 biomedicines-13-00439-f006:**
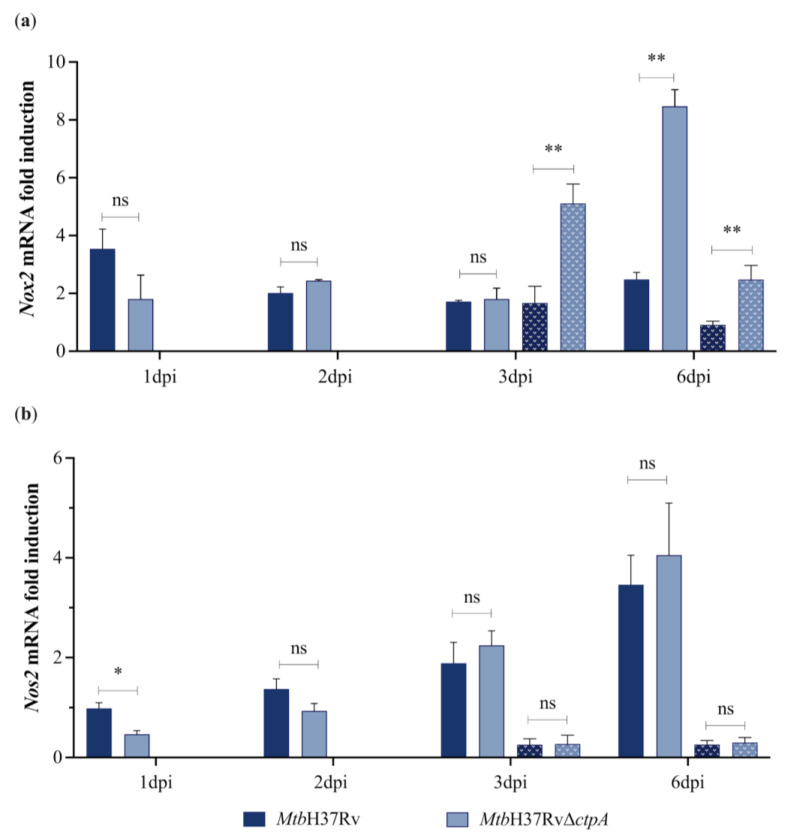
Gene transcription of NADPH oxidase 2 and iNOS in Mtb-infected MH-S macrophages. The gene expression of *Nox2* gene (**a**) and *Nos2* (**b**) in alveolar macrophages infected with *Mtb*H37Rv and *Mtb*H37RvΔ*ctpA* at MOI 1:4 in the absence (solid bar) or presence of 50 μM CuSO_4_ (dotted bar) were quantified by RT-qPCR. Relative transcription levels were calculated using the Pfaffl method concerning uninfected cells using the *gapdh* gene as a house keeping gene. Presented values correspond to the mean ± SEM of the triplicate of four independent experiments. Asterisks indicate statistical significance (two-tailed unpaired *t*-test comparison, ns *p* > 0.05; * *p* < 0.05; ** *p* < 0.01).

**Figure 7 biomedicines-13-00439-f007:**
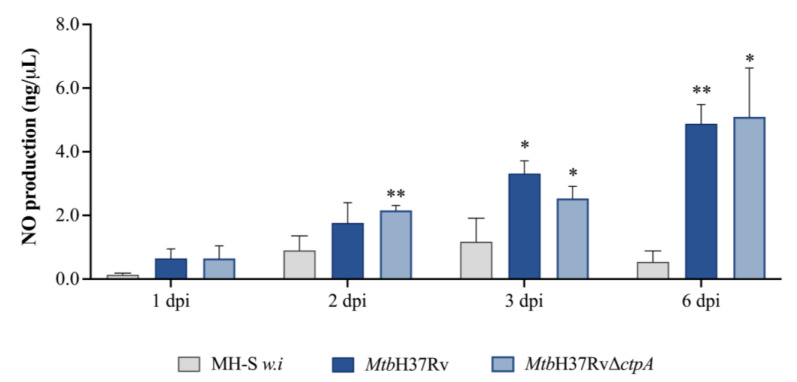
Nitric oxide (NO) production in MH-S macrophages infected with Mtb. Quantification in supernatants of NO produced by 1x10^5^ alveolar macrophages (MH-S) non-infected (*w.i*) or infected at 1:3 MOI with *Mtb*H37Rv or *Mtb*H37RvΔ*ctpA* was determined by Griess reagent as total nitrite measured at 540 nm [36], using a NaNO_2_ calibration curve between 0.33 and 119 ng/μL Asterisks above the bars indicate the two-tailed *t*-test (unpaired) comparison test regarding NO production in uninfected MH-S cells (* *p* < 0.05; ** *p* < 0.01).

**Figure 8 biomedicines-13-00439-f008:**
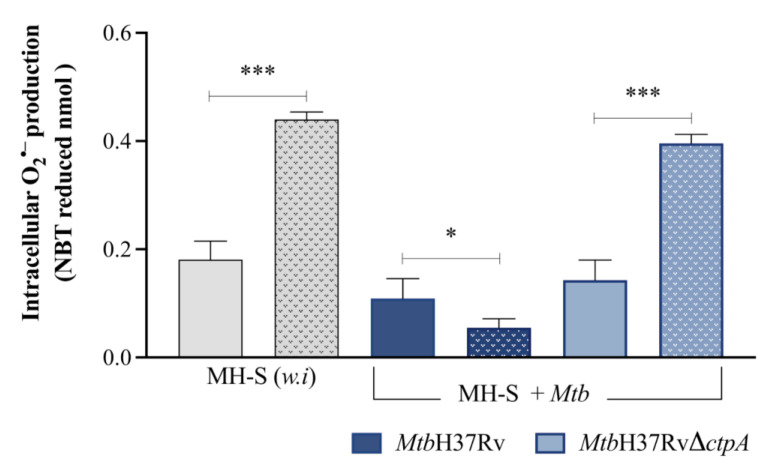
Intracellular ROS production from Mtb-infected MH-S *cells***.** Intracellular superoxide anion production by MH-S macrophages infected with *Mtb*H37Rv or *Mtb*H37RvΔ*ctpA* (MOI 1:3), in the absence (solid bar) or presence of the inducer PMA (900 ng/μL) (dotted bar), was measured using the modified colorimetric NBT assay [41,42]. Stimulation with PMA in uninfected (*w.i*) macrophages is the positive control of ROS production. As a negative control, the cells were incubated with 45 μg/mL SOD, but the levels obtained were below the detection limit. Asterisks indicate the two-tailed (unpaired) *t*-test comparison nmol NBT without PMA stimulation (* *p* < 0.05; *** *p* < 0.001).

**Figure 9 biomedicines-13-00439-f009:**
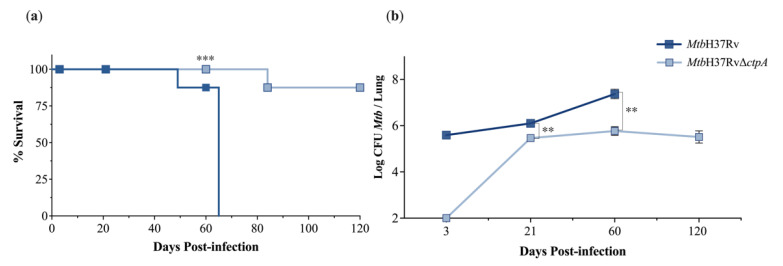
Survival and lung bacillary burdens in BALB/c mice infected with either ctpA mutant or parental MtbH37Rv strains. (**a**) Almost all mice infected with mutant *ctpA* strain survived, while all animals died after 60 days of infection with the parental strain. (**b**) Groups of four animals were euthanized at the indicated days and lungs were used for the determination of live bacillary burdens by the quantification of CFU. At 3 dpi, the number of CFU in animals infected with the mutant strain was so low that it was below the limit of detection (2 Log/CFU). Asterisks indicate the significant difference by two-tailed unpaired *t* test (** *p* < 0.01; *** *p* < 0.001).

**Figure 10 biomedicines-13-00439-f010:**
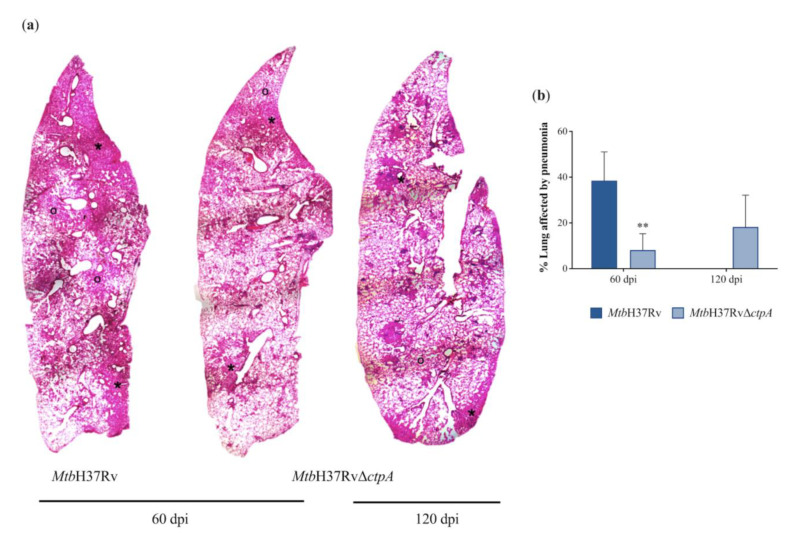
Pneumonic areas in the lungs of Mtb-infected mice. (**a**) Representative low power micrographs of the lungs stained with hematoxylin and eosin (H&E) that show the extensive consolidated pneumonic (asterisks) and alveolitis (o) areas in the lungs of *Mtb*H37Rv, while *Mtb*Δ*ctpA*-infected mice at 60 and 120 days post-infection showed lesser tissue damage. (**b**) This was confirmed by the measurement of pneumonia by automated morphometry. Asterisks above the bars show a significant difference between wild-type and mutant strain (** *p* < 0.01). (Micrographs 25× magnification). Software Leica Application Suite v4.0.

## Data Availability

The data presented in this study are available on request from the corresponding author.

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
