# Peer review of "The Plasma Membrane P-Type ATPase CtpA Is Required for Mycobacterium tuberculosis Virulence in Copper-Activated Macrophages in a Mouse Model of Progressive Tuberculosis"

_biomedicines, 2025, doi:10.3390/biomedicines13020439_

Round 1
Reviewer 1 Report
Comments and Suggestions for Authors
Major comments:
The results shall be presented in a more organized and succint way, as indicated in the comments below.
In the in vivo experiment, the differences observed at day 3 p.i. pose a major problem in interpreting the results at later time points. It is indeed possible that the observed differences at day 3 are responsible for the different outcomes in terms of survival, pathology and CFUs at later time points. But for this very reason it is not possibile to make any consideration on the role of ctpA during the acute and chronic phase of infection. In fact, to make such comparison, CFUs at day 1 or at day 3 shall be similar in the mutant and parental strains. This is a major issue that has significant impact on the conclusion.
Specific comments:
MH-S experiments with Mtb. Line 151: it is not clear whether the RPMI with antibiotics was used to wash Mtb infected cells or also used to incubate infected cells. Why antibiotics were added anyway?
Figure 2 and associated text. Intracellular Mtb shall be measured as CFUs/number of cells rather than CFUs/ml. This shall be corrected. I suggest to merge the results shown in figure 2 (A & B) in a single graphic, using a timeline. In this way, it is easy to observe the differences of the two mutants in presence or absence of CuSO4. Such a graphical representation may require less text to present the data.
Paragraph 3.3 is quite confusing. Again using two charts in fig 3 and figure 4 to present data with or without CuSO4 does not help the reader. Showing charts with different Y-axis is confusing. Line 318-320: higher level of IL1B only in presence of CuSO4!
Line 320-322: It transpires that higher levels of Il-12 and TNF are observed only in the absence of CuSO4 (see figure) and this must be indicated.
Line 322-324: stating that “ctpA deletion does not affect the immunogenicity of mtb in vitro” seems in contrast with the results presented. Indeed, deletion of ctpA affects immunogenicity.
Line 336: this is confusing. At which time point this reduction is observed? It seems the contrary.
All the paragraph is very difficult to follow. A better graphical representation of the results in merged charts will certainly help the readers and reduce the wording in the text.
Figure 6: why this experiment was not performed with the addiction of CuSO4?
Results on the in vivo experiments require extensive revision. Data shown in figure 8B shall be in Log CFU. Non need to use too many words or calculate the relative increase. Rather, present the data in an effective graphical format on the chart (timeline).
Highlight the fact that at day 3 p.i. mice infected with the ctpA mutant could not be detected. This is very important.
Reviewer 2 Report
Comments and Suggestions for Authors
In this manuscript, the authors investigated the role of the 20 copper-transporting P-type ATPase CtpA in Mtb virulence and found that MH-S cells infected with wild-type (MtbH37Rv) or the mutant strain (MtbH37RvΔctpA) 25 showed no difference in Mtb bacterial load. However, the same macrophages under copper activa- 26 tion (50µM CuSO4) showed impaired replication of the mutant strain. Furthermore, the mutant 27 MtbΔctpA strain showed an inability to control reactive oxygen species (ROS) induced by PMA ad- 28 dition during MH-S infection.
I have some concern about this investigation. The authors should clearly differentiate their study from previous.
The study previous studies already confirmed the role of CtpA and showed that it is non-essential for Mtb.
https://pmc.ncbi.nlm.nih.gov/articles/PMC6593218/
https://pmc.ncbi.nlm.nih.gov/articles/PMC5241402/
The word indeed used in different places may be suitable. There are some other phrases where the correct technical words is the responsibility of authors.
In methodology section 2.1, duration of MGIT tubes for culture should be mentioned.
How ΔctpA mutant was created? A brief description may be added.
A flowchart methodology may be added for better understanding.
Line 149, MOI of 1:4? Elaborate it.
Line 150, RPMI? The authors must write the full form of abbreviations while writing for the first time.
Line 159 [NF-α, and IL-12 concentrations in infected MH-S cell culture supernatants], how the the supernatant was obtained?
Line 169, NaNO2? Is it correct? How this curve was collected?
Lines 186-187, Efficiency curves were constructed using cDNA serial dilutions (0.01-1000 ng/μL) from uninfected MH-S macrophages. The value may be checked again.
Line 216, thawed and sonicated (for how much time)
The serial dilution in whole manuscript should be of same format.
Line 429, The caption of figure 8 may be revised.
Line 218, 434 etc, the word pneumonia has been used. Is it suitable this investigation where infectious agent if Mtb?
Line 539, CtpA plays a role in detoxifying ROS (need reference and a short mechanism)
Round 2
Reviewer 1 Report
Comments and Suggestions for Authors
Item 2:
The answer provided by the authors partly addresses the issues raised in my comment. Indeed, given the similar inoculum, the lower bacterial burden at day 3 for the mutant indicates an impaired ability to replicate in vivo in these three days or an an increased susceptibility of the mutant to the early events taking place in the alveoli (higher susceptibility to alveolar macrophages killing?). Does the attenuation last also at later time points? If, for example, the parental strain is inoculated to reach a CFU/lung at day 3 like that observed for the mutant, the growth curve in vivo would be always lower than the parental strain starting at day 3 at higher CFU/lung.
For these reasons, I consider more convincing the different slope of the curve between day 21 and day 60, that indicates an attenuation of the mutant, as properly outlined on lines 427-429.
Moreover, in figure 9 panel b, since the detection limit is 100 CFU, that is 2 LogCFU/lung, the Y axis shall properly report this limit and the point at day 3 for the mutant shall not be placed at 0.
Author Response
Dear
Reviewer Biomedicines –3392823
Thank you very much for taking the time to review this manuscript. Please find the detailed response below and the corresponding revisions /corrections highlighted/ in track changes in the re-submitted files
Comment 1:
The answer provided by the authors partly addresses the issues raised in my comment. Indeed, given the similar inoculum, the lower bacterial burden at day 3 for the mutant indicates an impaired ability to replicate in vivo in these three days or an an increased susceptibility of the mutant to the early events taking place in the alveoli (higher susceptibility to alveolar macrophages killing?). Does the attenuation last also at later time points? If, for example, the parental strain is inoculated to reach a CFU/lung at day 3 like that observed for the mutant, the growth curve in vivo would be always lower than the parental strain starting at day 3 at higher CFU/lung. For these reasons, I consider more convincing the different slope of the curve between day 21 and day 60, that indicates an attenuation of the mutant, as properly outlined on lines 427-429. Moreover, in figure 9 panel b, since the detection limit is 100 CFU , that is 2 LogCFU/lung, the Y axis shall properly report this limit and the point at day 3 for the mutant shall not be placed at 0.
Response 1: We agree with the reviewer comment, it seems that mutant bacilli are highly susceptible to be killed by alveolar macrophages during early infection (line 492-494; 497-500) as is commented in the discussion it seems that this highly susceptibility is because mutant bacteria are unable to survive against the oxidative injury, suggesting a potential role for CtpA in overcoming redox stress as suggested by our in-vitro results (line 534-543).
In regard to the infection with a lower concentration of the wild-type strain and the progression of the infection, we have conducted additional research in which we infected mice with a lower number of parental bacilli from the H37Rv strain, the lowest amount was 4000 bacilli which produce chronic infection with 100 000 UFC/lung or less until day 270 of infection, then a progressive increase was observed (Clin Exp Immunol 2002; 128:229–237). Thus, we agree that attenuation of the mutant bacilli is more convincing between day 21 and day 60 post-infection and as correctly was indicated. Furthermore, according to the reviewer suggestion at day 3 of the mutant in fig 9 panel b was now placed in 2 LogCFU instead of 0 in the Y axis, and the limit detection was added in the caption of the figure (line 439).
Finally, we look forward to having our paper published in “Biomedicines” and we extend our gratitude for your time and proper suggestions to make this manuscript stronger.
Dulce Mata-Espinosa, PhD
Sección de Patología
Instituto de Ciencias Médicas y Nutrición Salvador Zubiran, México
Carlos Y. Soto, Ph.D
Departamento de Química- Facultad de Ciencias
Universidad Nacional de Colombia